# Effectiveness of Cement Augmentation on Early Postoperative Mobility in Patients Treated for Trochanteric Fractures with Cephalomedullary Nailing: A Prospective Cohort Study

**DOI:** 10.3390/jpm12091392

**Published:** 2022-08-27

**Authors:** Yusuke Mochizuki, Norio Yamamoto, Tatsuya Fujii, Yosuke Tomita

**Affiliations:** 1Department of Orthopaedic Surgery, Kaneda Hospital, Okayama 719-3193, Japan; 2Department of Orthopedic Surgery, Miyamoto Orthopedic Hospital, Okayama 773-8236, Japan; 3Department of Epidemiology, Graduate School of Medicine, Dentistry and Pharmaceutical Sciences, Okayama University, Okayama 700-8558, Japan; 4Scientific Research Works Peer Support Group (SRWS-PSG), Osaka 541-0043, Japan; 5Department of Orthopaedic Surgery, Matsue Hospital, Tokyo 132-0025, Japan; 6Department of Physical Therapy, Faculty of Health Care, Takasaki University of Health and Welfare, Takasaki 370-0033, Japan

**Keywords:** cement augmentation, trochanteric fracture, intertrochanteric fracture, hip fracture, proximal femoral fracture, cumulated ambulation score, mobility, ambulance, ADL, pain

## Abstract

Fixation using cephalomedullary nails (CMNs) with additional cement augmentation (CA) was developed as a novel treatment option for the osteosynthesis of osteoporotic trochanteric fractures, though the effectiveness of CA on early postoperative mobility remains uncertain. This multicenter prospective cohort study aimed to estimate the effectiveness of CA on early postoperative mobility in patients with trochanteric fractures. We enrolled patients with femoral trochanteric fractures aged >60 years who were able to walk independently before the injury. The primary outcome was the postoperative 3-day cumulated ambulation score (CAS); the secondary outcome was the visual analog scale (VAS) pain score at rest and during movement on postoperative days 1–3. The outcomes of the patients treated using CMNs with or without CA were compared. Sixty-three eligible patients were categorized into CA (*n* = 32) and control (*n* = 31) groups. In univariate analysis, the CA group had significantly higher CAS values, lower VAS scores at rest on day 1 postoperatively, and lower VAS scores during movement on day 3. In multivariable linear regression analyses, the CA group had significantly higher CAS values (beta, 2.1; 95% confidence interval, 0.5 to 3.6; *p* = 0.01). The CA group had a negative adjusted beta value in their VAS scores during movement. This study indicated that CA was associated with a high CAS value in patients with geriatric trochanteric fractures. However, CA was not associated with pain reduction at rest and during movement during the initial postoperative days.

## 1. Introduction

Trochanteric fractures are common injuries among the elderly, and the incidence of these fractures continues to increase due to an aging society [1,2]. As the aging rate in patients with trochanteric fractures increases, more cases are complicated by severe osteoporosis [1,2]. Studies have focused on mechanical complications after internal fixation with cephalomedullary nails (CMNs), particularly on cut-out [3,4]. Enhanced mechanical stability with the cement augmentation method has been demonstrated in several biomechanical studies [5,6]. Therefore, fixation using CMNs with additional cement augmentation (CA) has been developed as a novel treatment option for the osteosynthesis of trochanteric fractures in osteoporotic bones [7,8,9,10].

The effectiveness of CA on mobility in the early postoperative period has not been investigated to date, despite the significance of the early performance of activities of daily living (ADL) in reducing postoperative complications and regaining ambulation after surgery [11]. A systematic review reported that the effectiveness of CA on functional outcomes was uncertain because only a few randomized control trials assessing the functional outcomes on CA had been performed [10].

Therefore, this study aimed to estimate the effectiveness of CA on early postoperative mobility in patients with trochanteric fractures. We specifically focused on the early postoperative period. We hypothesized that the enhanced stability with CA would reduce loading pain, allowing patients to achieve increased mobility and regain preoperative ADL performance in the early postoperative period and reduce perioperative complications.

## 2. Materials and Methods

### 2.1. Study Design and Setting

This is a multicenter prospective cohort study of patients with trochanteric fractures between February and December 2021 who were treated by two orthopedic surgeons at two general hospitals in Japan. We designed the study in accordance with the Strengthening the Reporting of Observational Studies in Epidemiology (STROBE) criteria [12]. This study was approved by the ethics committees of the hospitals (No. 1006). We registered the protocol before participant recruitment [13].

### 2.2. Patient Selection

The inclusion criteria were as follows: patients who underwent surgical treatment for first-time trochanteric fractures between February and December 2021, fracture types of the AO Foundation/Orthopedic Trauma Association (AO/OTA) classification 31A [14] (pertrochanteric fracture [31A1,2] and intertrochanteric fracture [31A3]), patients who were aged >60 years, and patients with the ability to walk independently prior to the injury (walking without aid or with a cane or walker). The exclusion criteria were as follows: pathological fractures, open fractures, use of a wheelchair before the injury, severe heart or lung diseases, history of allergy to cement, multiple lower extremity trauma, postoperative inability to bear weight, and postoperative medical complications making it difficult to leave the bed (pneumonia, heart failure, etc.). The indications for CA depended on the preference and judgment of the patient and surgeon, and were based on the indication criteria followed in Japan.

### 2.3. Surgical Procedures and Rehabilitation Plan

All patients underwent surgery under general anesthesia. First, we performed a closed reduction on a traction table under fluoroscopic guidance. When adequate fracture reduction was not achieved, procedures to achieve the anteromedial cortical reduction using a lag screw incision or additional mini-open anterolateral incision were performed [15]. We then fixed the fracture with a Trochanteric Femoral Nail Advanced (TFNA) perforated spiral blade (DePuy Synthes, Paoli, PA, USA). The patient was allowed full weight-bearing immediately after surgery under medical staff guidance (doctors, rehabilitation staff, and nurses) according to their pain severity and medical condition.

### 2.4. Main Exposure

The main exposure in this study was CA. In patients undergoing CA, Traumacem (DePuy Synthes, Paoli, PA, USA), a polymethyl methacrylate (PMMA) cement, was injected into the blade (3–6 mL) under fluoroscopic control in accordance with the manufacturer’s recommendations (Figure 1). We did not use contrast material to check the presence of perforation by guide pin insertion prior to cement injection. We sufficiently confirmed that the cement did not leak into the hip joint under fluoroscopy.

### 2.5. Outcomes

The primary outcome was the total Cumulated Ambulation Score (CAS) on postoperative days 1, 2, and 3. The CAS is a valid and reliable assessment tool for evaluating a patient’s mobility by observing the following three basic movements [16]: (1) getting in and out of bed, (2) sitting and rising from a chair (with armrests), and (3) indoor walking (with or without a walking aid). The CAS is superior to other measures in assessing the mobility of patients with hip fractures [17,18]. Each basic movement is graded from 0 to 2 (a higher score means better performance), and CAS values can range from 0 to 6 in 1 day. The total CAS values for 3 postoperative days ranged from 0 to 18. The rehabilitation staff, who had sufficient knowledge of CAS and were blind to the research protocol, scored patients on the same day.

The secondary outcome was the pain score measured using the visual analog scale (VAS) at rest and during movement on postoperative days 1–3. The Barthel index (BI), ambulance ability (wheelchair, parallel bars, walker, cane, or walking alone), and return to pre-fracture ambulatory level (RPAL) were evaluated one week postoperatively. We investigated the postoperative complication classification system (Sink classification [19]) and other medical complications that occurred within 1 week postoperatively (coronary artery disease, gastrointestinal bleeding, acute renal failure, delirium (diagnosed based on the confusion assessment method) [20], stroke, venous thrombosis, pneumonia, urinary tract infection, wound infection, pressure ulcer, perioperative blood transfusion, or death). Specific adverse events related to the CA, such as cement allergy and cement leakage in the hip joint, were included.

### 2.6. Preoperative Variables

The preoperative variables were as follows: patient demographics (age, sex, height, weight, and body mass index (BMI)), Charlson comorbidity index (CCI) [21] including dementia, American Society of Anaesthesiologists classification (ASA), pre-fracture ambulatory level, preoperative blood test values including hemoglobin and albumin [22], pre-injury residence (single, co-residence, institutional, and hospital), treatment for osteoporosis, fracture characteristics based on AO/OTA classification [14], and preoperative waiting period (days from hospital admission until surgery).

### 2.7. Postoperative Variables

The postoperative variables included reduction quality of fracture, the blade position, surgical time (min), and intraoperative blood loss (cc).

The overall reduction quality was evaluated according to the Baumgaertner criteria on a postoperative radiograph [23]. The anteromedial cortex fracture reduction quality was classified into two types: adequate (extramedullary and anatomical) or inadequate (intramedullary) [24,25]. We assessed the quality of blade placement in the femoral head with the tip–apex distance (TAD) [23] and position [26].

### 2.8. Statistical Analyses

We performed a sample size calculation for two-group comparisons. Based on previous studies using CAS as an outcome variable in acute hip fractures [18], a sample size of 42 patients (21 patients per treatment group) is needed for this study to have 80% power to detect a 2-point mean difference in CAS scores with a type I error of 5%. Therefore, we set a sample size of 50 patients (25 patients per treatment group) to accommodate for patient dropouts.

The normal distribution of the data was tested using the Kolmogorov–Smirnov test. Continuous data with normal distribution were presented as means (standard deviations (SD)) and compared using a t-test between two comparisons. Those with non-normal distribution were presented as medians and interquartile ranges and compared using the Mann–Whitney U test. Categorical data were presented as a proportion of cases and compared using Fisher’s exact test or the χ^2^ test, as appropriate.

Multivariable linear regression analysis was performed only for the outcomes that met with the assumption. We selected possible confounding factors (age, dementia, and CCI) based on previous studies [22,27]. The goodness-of-fit was assessed using the adjusted R squared and *p*-values. A *p* of <0.05 was considered statistically significant. All statistical analyses were performed using Stata SE version 17.0 (StataCorp, College Station, TX, USA).

## 3. Results

After considering the inclusion and exclusion criteria, 63 eligible patients were identified (Figure 2). The study included 63 patients with a mean age of 87.0 (65–104) years. In total, 49 (77.8%) women and 14 (22.2%) men were included. No data were missing.

Table 1 summarizes the patients’ baseline characteristics and preoperative radiographic findings between the CA and control groups. The two groups did not differ significantly, except in terms of gender. The control group had significantly more male patients than the CA group (*p* = 0.01).

No significant difference was noted in postoperative radiographic findings and intraoperative data (Table 2). No specific adverse events related to CA were observed. Table 3 summarizes the postoperative outcomes of the two groups. The CAS values were significantly higher in the CA group than in the control group (*p* = 0.004). The VAS scores at rest on day 1 and the VAS scores during movement on day 3 were significantly lower in the CA group than in the control group (*p* = 0.003 and 0.004, respectively). No significant differences in postoperative complications were noted (*p* = 0.212). The patients with CA had higher BIs than those without, although the difference was not significant (*p* = 0.247).

In the multiple linear regression analyses, the patients with CA had significantly higher CAS values than those without (beta, 2.1; 95% confidence interval (CI), 0.5 to 3.6; *p* = 0.01) (Table 4). The patients with CA had a negative adjusted beta value in their VAS scores during movement on days 2 and 3 (Appendix A).

## 4. Discussion

This study was a multicenter prospective cohort study to assess the effectiveness of CA on early postoperative ADL scores in patients with trochanteric fractures. The patients with CA had significantly higher CAS values than those without (after adjusting for confounding factors). However, CA did not reduce pain at rest and during movement (after adjusting for confounding factors), nor did it improve Barthel index values at 1 week postoperatively or reduce postoperative medical complications significantly.

CA had clinical effectiveness on high CAS values in patients with geriatric trochanteric fractures. Even after adjusting for dementia, sex, and age, which were unmodifiable risk factors in the clinical setting, CA is an affecting factor that causes clinically and statistically significant differences to facilitate CAS enhancement [18,27]. These better results with CA on postoperative mobility are consistent with the findings of previous studies that show a high proportion of RPAL [28] and weight-bearing [29] in patients with CA. As reported in earlier studies [9,28,29,30,31,32] (Table 5), we speculate that CA enhances a patient’s initial mechanical stability, leading to an improved ADL score due to sufficient stability in the early postoperative period.

Our results showed that CA did not reduce pain shortly after surgery. Mechanical stability enhanced with CA may not reduce pain shortly post-surgery, although most surgeons expect this positive effect. There are conflicting studies regarding the effectiveness of CA on pain reduction (Table 5). Two studies reported significant pain reduction at postoperative week 2 and at 12 months [32,33], but another study showed no such effect at 6 months postoperatively [31]. The inconsistent results may be partly due to the fact that these previous observational studies did not adjust for confounders in their statistical models. In our study, we included several possible confounders such as age, dementia, and CCI which increased the statistical models’ precision. Therefore, as there was no clear difference in early postoperative pain between patients with and without CA in our study, this may provide more robust results than those of previous studies (if the confounders are accounted for). Indeed, further investigations over time with more assessment time points are needed.

### 4.1. Strengths

First, to the best of our knowledge, this was the first prospective cohort study to demonstrate the effectiveness of CA on early postoperative ADL scores for elderly patients with trochanteric fracture. Second, the results revealed the effectiveness of CA on CAS values after adjusting confounding factors. The study successfully compensated for the scarce rehabilitation evidence in previous studies because we assessed early postoperative mobility conditions using CAS values. Early mobility improvement in CA is beneficial to the patient’s activity, and it also reduces the socioeconomic burden for medical staff and families. Targeting the benefit of CA represents a significant change in practice because the incidence of geriatric trochanteric fractures is increasing due to an aging population, and it has a heavy socioeconomic burden [1,2]. The mechanical strength of CA has been recognized as clear evidence; hence, the results of this study may influence existing trochanteric fracture care guidelines and policies on rehabilitation.

### 4.2. Limitations

First, the measurement bias on outcome scoring by some medical staff may result in non-differential misclassification in the two groups, leading to minimal effects on the outcomes. Second, the eligible sample size of 64 patients was small, leading to an underpowered analysis being used to detect the differences in the other outcomes. Third, we did not evaluate the number of rescue analgesics used and other anesthesia parameters for postoperative pain. The use of analgesics was dependent on each patient. Some patients were also originally taking analgesics regularly. The pain scores in patients with dementia may have had measurement bias due to invalid and less reproducible assessments. Fourth, we did not evaluate bone mineral density (BMD) due to too many missing values. In the biomechanical study, CA enhanced torque force, especially in osteoporotic (low BMD) specimens [34]. BMD is an important factor in assessing the effect of CA. Fifth, we could not evaluate medical costs and clinical outcomes for a longer duration (3–12 months). Previous studies have shown that CA was significantly associated with an increased RPAL at 12 months [28], although the RPAL at 1 week in our study did not differ significantly. Sixth, there is a lack of external validity because of our inclusion criteria and data from only two general hospitals in Japan were used. It remains unclear whether the results of the study can be generalized to other countries with different patient characteristics and healthcare systems. Lastly, this study is a prospective cohort study with some limitations. Further well-designed, randomized controlled trials are needed to clarify the effectiveness of CA.

## 5. Conclusions

This prospective cohort study indicated that CA was associated with high CAS values in patients with geriatric trochanteric fractures. However, CA was not associated with pain reduction at rest and during movement during the early postoperative days.

## Figures and Tables

**Figure 1 jpm-12-01392-f001:**
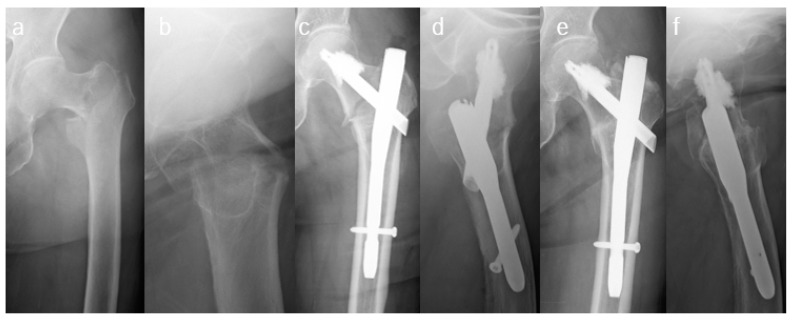
(**a**,**b**) Preoperative radiographs showing the left trochanteric fracture in an 87-year-old female. (**c**,**d**) Immediate postoperative radiographs showing the fracture fixed with TFNA using cement augmentation. (**e**,**f**) Postoperative 9-months radiographs indicating that the fracture has healed.

**Figure 2 jpm-12-01392-f002:**
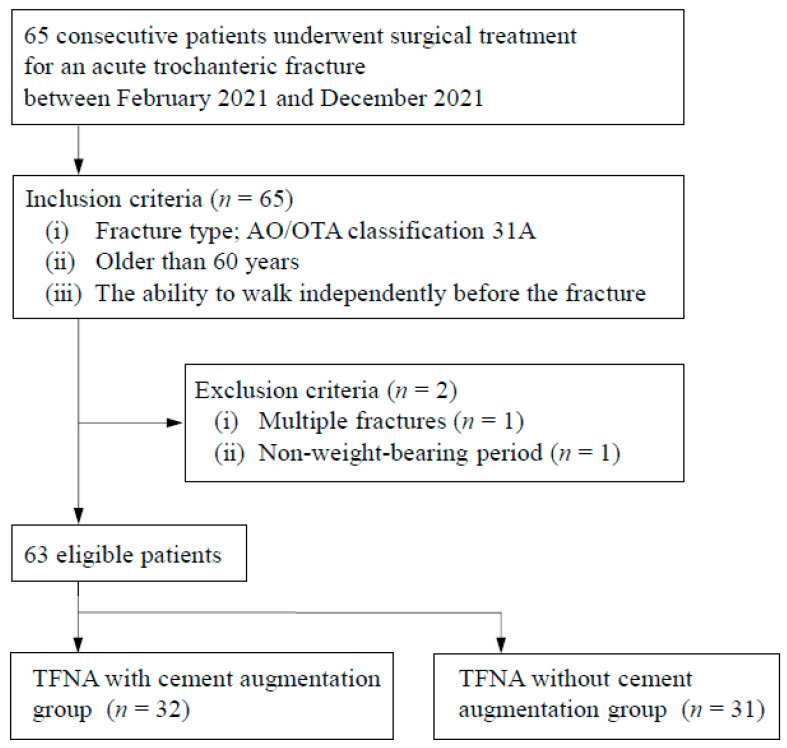
Patient flowchart. TFNA, trochanteric femoral nail advanced; AO/OTA, AO Foundation/Orthopedic Trauma Association.

**Table 1 jpm-12-01392-t001:** Perioperative data of patients treated using TFNA with and without cement augmentation.

	Total (*n* = 63)	Cement Augmented Group (*n* = 32)	Control Group (*n* = 31)	*p-*Value
Mean age, years (range)	87.0 (65–104)	87.0 (65–99)	87.0 (66–104)	0.98
Men, *n* (%)	14 (22.2)	3 (9.4)	11 (35.5)	0.01
Height, cm (SD)	150.6 (9.3)	149.2 (8.9)	152.1 (9.7)	0.22
Weight, kg (SD)	47.5 (9.6)	46.9 (9.8)	48.2 (9.5)	0.59
BMI, kg/m^2^ (SD)	20.8 (2.9)	21.0 (3.2)	20.7 (2.6)	0.71
ASA classification 1, *n* (%)	13 (20.6)	7 (21.9)	6 (19.4)	1.00
ASA classification 2, *n* (%)	50 (79.4)	25 (78.2)	25 (80.6)	1.00
Charlson comorbidity index, median (IQR)	2 (0–2)	2 (0–2)	2 (0–2)	0.51
Preoperative laboratory value				
Hemoglobin, g/dL (SD)	10.7 (1.8)	10.6 (1.6)	10.9 (2.1)	0.61
Albumin, g/dL (SD)	3.3 (0.5)	3.3 (0.4)	3.3 (0.5)	0.91
Pre-fracture ambulatory level				
IA without gait aids, *n* (%)	32 (50.8)	17 (53.1)	15 (48.4)	0.70
Pre-fracture resistance (single- and co-residence), *n* (%)	52 (82.5)	25 (78.2)	27 (87.1)	0.34
Treatment for osteoporosis, *n* (%)	6 (9.5)	4 (12.5)	2 (6.5)	0.41
Dementia, *n* (%)	37 (58.7)	20 (62.5)	17 (54.8)	0.54
Fracture type A1, *n* (%) (A1.2/1.3)	48 (76.2)	23 (71.9)	25 (80.6)	0.41
Fracture type A2, *n* (%) (A2.2/2.3)	15 (23.8)	9 (28.1)	6 (19.4)	0.41
Time to surgery, day (SD)	3.3 (1.8)	3.2 (1.9)	3.4 (1.8)	0.45

TFNA, trochanteric femoral nail advanced; ASA, American Society of Anesthesiologists; BMI, body mass index; IA, independent ambulation; SD, standard deviation; IQR, interquartile range.

**Table 2 jpm-12-01392-t002:** Postoperative radiographic findings and intraoperative data outcomes of the patients treated using TFNA with and without cement augmentation.

	Total (*n* = 63)	Cement Augmented Group (*n* = 32)	Control Group (*n* = 31)	*p*-Value
TAD, mm (SD)	20.2 (4.8)	21.1 (4.5)	19.3 (5.1)	0.15
Center-center blade position, *n* (%)	55 (87.3)	29 (90.6)	26 (83.9)	0.42
Adequate reduction with AP Xp, *n* (%)	63 (100)	32 (100)	31 (100)	1.00
Adequate reduction with lateral Xp, *n* (%)	58 (92.1)	30 (93.8)	28 (90.3)	0.61
Baumgaertner criteria, good, *n* (%)	60 (95.2)	30 (93.8)	30 (96.8)	0.57
Surgical time, min (SD)	47.5 (21.7)	46.0 (22.7)	49.0 (20.9)	0.59
Intraoperative blood loss, cc, median (IQR)	50 (45–50)	50 (47.5–100)	50 (45–50)	0.52

TFNA, trochanteric femoral nail advanced; TAD, tip–apex distance; SD, standard deviation; IQR, interquartile range.

**Table 3 jpm-12-01392-t003:** Comparison of postoperative outcomes between patients treated using TFNA with and without cement augmentation.

	Total (*n* = 63)	Cement Augmented Group (*n* = 32)	Control Group (*n* = 31)	*p*-Value
CAS at day 1–3, median (IQR)	4.0 (2–6)	5.0 (3.8–6.3)	3.0 (1.0–4.5)	0.004
VAS at rest day 1, median (IQR)	2.0 (0–5.3)	0.09 (0–2.3)	4.6 (0–7.3)	0.003
VAS at rest day 2, median (IQR)	0 (0–2.2)	0 (0–2.0)	0.7 (0–2.7)	0.183
VAS at rest day 3, median (IQR)	0 (0–1.9)	0 (0–0.71)	0 (0–3.0)	0.060
VAS during movement day 1, median (IQR)	8.0 (5.9–9.3)	8.0 (6.0–8.3)	8.0 (5.7–10)	0.241
VAS during movement day 2, median (IQR)	5.7 (4.0–8.0)	5.0 (4.0–7.0)	6.0 (4.3–8.0)	0.225
VAS during movement day 3, median (IQR)	5.0 (3.0–7.2)	4.0 (2.3–5.3)	7.0 (3.6–8.4)	0.004
Barthel index at 1 week, median (IQR)	40 (15–52.5)	45.0 (20–51.3)	35.0 (10–52.5)	0.247
RPAL at 1 week, *n* (%)	2 (3.2)	2 (6.3)	0 (0)	0.150
Sink classification 1/2, *n* (%)	50/13 (79.4)	23/9 (71.9)	27/4 (87.1)	0.136
Complications during postoperative week 1, *n* (%)	42 (66.7)	19 (59.4)	23 (74.2)	0.212

TFNA, trochanteric femoral nail advanced; TAD, tip–apex distance; CAS, cumulated ambulation score; VAS, visual analog scale; SD, standard deviation; IQR, interquartile range; RPAL, return to pre-fracture ambulatory level.

**Table 4 jpm-12-01392-t004:** Multivariable linear regression analysis for CAS.

	Adjusted R2	Beta (95% CI)	Adjusted Beta (95% CI)	*p*-Value
	0.121			0.01
Constant		3.95 (−5.66 to 13.55)		0.42
Age		0.01 (−0.11 to 0.12)	0.01 (−0.26, 0.28)	0.95
CCI		−0.22 (−0.75 to 0.31)	−0.12 (−0.39, 0.16)	0.40
Dementia		−0.55 (−1.51 to 0.42)	−0.17 (−0.48, 0.13)	0.26
Cement augmentation		2.05 (0.51 to 3.59)	0.33 (0.08, 0.59)	0.01

CAS, cumulated ambulation score; CCI, Charlson comorbidity index; CI, confidence interval.

**Table 5 jpm-12-01392-t005:** Literature review of clinical physical activity evaluation of trochanteric fracture cases treated using cephalomedullary nails with and without CA.

Authors	Study Design	CA Cases (*n*)	Control Cases (*n*)	Implant	Timing of Evaluation (Post Operation)	Clinical Outcomes
Dall’Oca et al. (2010) [30]	RCT	40	40	Gamma nail 3	1,3,6,12 months	Harris hip score; without a certain tendency
Kammerlander et al. (2018) [9]	RCT	85	115	PFNA	3,6,12 months	There were no significant differences in time taken to walk 3 m and RPAL.
Kim.et al. (2018) [31]	Retrospective cohort study	40	42	PFN	6 months	The CA group had higher Harris hip scores and lower VAS pain scores.
Keppler et al. (2021) [29]	Prospective cohort study	24	24	PFNA	5 days	The CA group had higher Barthel index values and loading rates.
Mitsuzawa et al. (2021) [32]	Prospective cohort study	9	9	TFNA	2 weeks	The CA group had lower VAS scores in full load walking and higher Parker mobility scores.
Kulachote et al. (2021) [28]	Retrospective cohort study	68	67	PFNA	1 year	The CA group had higher RPALs.
Our study (2022)	Prospective cohort study	32	32	TFNA	1–3 days, 1 week	The CA group had higher CASs after multiple linear regression analysis.

CA, cement augmentation; RCT, randomized controlled trial; VAS, visual analog scale; RPAL, return to prefracture ambulatory level; PFNA, proximal femoral nail antirotation; TFNA, trochanteric femoral nail advanced; CAS, cumulated ambulation score.

## Data Availability

Not applicable.

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
