# Peer review of "Effectiveness of Cement Augmentation on Early Postoperative Mobility in Patients Treated for Trochanteric Fractures with Cephalomedullary Nailing: A Prospective Cohort Study"

_jpm, 2022, doi:10.3390/jpm12091392_

Round 1
Reviewer 1 Report
Title:
Ok, although my suggestion is to shorten the title.
Abstract:
Ok.
Introduction:
Line 37: “Trochanteric fractures are the most frequently encountered fractures in clinical settings, 37 and the number of these fractures continues to increase owing to an aging society”
This sentence is incorrect. I would edit this sentence. Such as “common injuries among the elderly population”
Additionally, the term “trochanteric fractures” needs clarification. Did the authors mean inter\per trochanteric fractures? please clarify and correct throughout the text.
Methods:
Line 72: “the ability to walk independently prior to the injury (walking without aid or with a cane or a walker)
This inclusion criteria are quite broad, and may affect and prevent a firm conclusion regarding the early postoperative mobility. Please clarify and elaborate. additionally , add to the limitation section.
Line 76: elaborate the indication for CA.
Line 79: “63 eligible patients were identified (Figure 1).” Is more suitable for the result section.
Line 83: Surgical procedures and rehabilitation plan-
Please elaborate regarding anesthesia. Were all patients undergoing the surgery under general anesthesia? Were there cases of regional anesthesia, as this may affect the early postoperative rehab.
Blade was used in all cases? or there were cases with a screw? please clarify.
Line 93: Main exposure
Did you used a contrast material prior to cement injection to verify there is no leakage? Please elaborate.
Line 108: “The secondary outcomes were pain scores measured using the visual analog scale (VAS) at rest and during movement on postoperative days 1–3.”
How can the VAS be attributed to the use of cement without considering the type of anesthesia, the number of analgesics consumed and other anesthesia parameters that can affect postoperative pain? Please explain or elaborate, if this was not assessed please include in the limitations of the study.
Results:
Line 159: “The study included 64 patients with a mean age of 87.0 (65–104) years. In total, 49 (77.8%) 159 women and 14 (22.2%) men were included.”
49+14=63…
Table 1 – do you have an explanation why the average time from admission to surgery was more the 48 h?
Were AO type 31A3 were excluded prior to the study? If so, add to the exclusion criteria.
Line 189: “Patients with CA had higher Bis than those without”
A typo? Bis? Correct to BI
Discussion:
Please comment, or discuss, why do you think, even though there is no difference in the VAS between groups, the cement still affects the early postoperative function?
Please discuss the further cost of CA. for example, in the institution where I work, there is an allocation for a limited number of cements uses, for specific indications, since the cost of using cement is significantly higher than a CMN alone.
Limitations:
Address the lack of reference to the type of anesthesia affects the early postoperative function
Line 277: “Lastly, although there were several 277 limitations, including unmeasured factors, in the present study, our results will draw 278 attentions to the potential impact of CA on postoperative mobility. Ultimately, a well-279 designed study with a large sample size and long follow-up period is warranted to clarify 280 the effectiveness of CA.”
This is not a limitation.
Figures:
For the sake of the readers, I suggest adding a figure of a radiograph or intraoperative fluoroscopy that demonstrate the use CA.
Author Response
Response to Reviewer 1 Comments:
Title: Ok, although my suggestion is to shorten the title.
Response: We thank Reviewer 1 for these comments. We followed the STROBE Statement [1]; Checklist of items that should be included in reports of cohort studies. The STROBE states “Indicate the study’s design with a commonly used term in the title”. We have tried to shorten the title in the first version of manuscript. However, if further shortening is needed, please let us know.
[1] von Elm, E.; Altman, D.G.; Egger, M.; Pocock, S.J.; Gøtzsche, P.C.; Vandenbroucke, J.P.; et al. The Strengthening the Reporting of Observational Studies in Epidemiology (STROBE) Statement: Guidelines for Reporting Observational Studies. Int. J. Surg. 2014, 12, 1495–1499.
Introduction:
Line 37: “Trochanteric fractures are the most frequently encountered fractures in clinical settings, and the number of these fractures continues to increase owing to an aging society”
This sentence is incorrect. I would edit this sentence. Such as “common injuries among the elderly population”
Response: We agree with your comment as trochanter fractures are one of the most common injuries among the elderly. Therefore, we have revised the sentence as follows.
Line 37:
“Trochanteric fractures are common injuries among the elderly, and the incidence of these fractures continues to increase due to an aging society.”
Additionally, the term “trochanteric fractures” needs clarification. Did the authors mean inter\per trochanteric fractures? please clarify and correct throughout the text.
Response: As suggested, the definition of fracture classification is important. In the methods section, we have included the AO Foundation/Orthopedic Trauma Association (AO/OTA) classification 31A. It means that we included both pertrochanteric (31A1,2) and intertrochanteric fractures (31A3). We have edited the sentence as follows.
Line 70:
The inclusion criteria are as follows: patients who underwent surgical treatment for first-time trochanteric fractures between February and December 2021, fracture types of AO Foundation/Orthopedic Trauma Association (AO/OTA) classification 31A [14] (pertrochanteric fracture [31A1,2] and intertrochanteric fracture [31A3]), aged >60 years, and the ability to walk independently prior to the injury (walking without aid or with a cane or walker).
Methods:
Line 72: “the ability to walk independently prior to the injury (walking without aid or with a cane or a walker)
This inclusion criteria are quite broad, and may affect and prevent a firm conclusion regarding the early postoperative mobility. Please clarify and elaborate. additionally , add to the limitation section.
Response: We thank Reviewer 1 for raising an important point. We adopted a wide inclusion criteria to maximize the external validity of the study. Our inclusion/exclusion criteria allowed us to infer the effects of CA on a broad range of patients with independent mobility before the injury, which is the patient population that could benefit from an early regain of mobility in the clinical setting. However, we agree with Reviewer 1 that the wide inclusion criteria limited our analysis in terms of which specific population could benefit from the CA (e.g., severe osteoporosis, unstable fracture type, etc.). Therefore, we have added these points to the study limitation as suggested by the reviewer.
Line 275:
Fifth, there is a lack of external validity because of our inclusion criteria and data from only two general hospitals in Japan were used. It remains unclear whether the results of the study can be generalized to other countries with different patient characteristics and healthcare systems.
Line 76: elaborate the indication for CA.
Response: In Japan, available CMN with CA is only one type now, which is TFNA (Depuy Synthes). In Japan, the indication for CA in patients with trochanteic fractures is osteoporotic trochanteic fracture [1,2]. The contraindications were as follows; severe heart or lung diseases and allergy to cement. We have edited the following sentence.
Page 3:
The exclusion criteria are as follows: pathological fractures, open fractures, use of a wheelchair before the injury, severe heart or lung diseases, history of allergy to cement, multiple lower extremity trauma, postoperative inability to bear weight, and postoperative medical complications making it difficult to leave the bed (pneumonia, heart failure, etc.). The indications for CA depend on the preference and judgment of the patient and surgeon, and are based on the indication criteria followed in Japan.
[1] https://www.info.pmda.go.jp/downfiles/md/PDF/340216/340216_30200BZX00222000_A_01_01.pdf (In Japanese, accessed August 12, 2022)
[2]
http://synthes.vo.llnwd.net/o16/LLNWMB8/INT%20Mobile/Synthes%20International/eIFU-EMEA/532126/532126AKeng.pdf (Accessed August 12, 2022)
Line 79: “63 eligible patients were identified (Figure 1).” Is more suitable for the result section.
Response: As suggested, we have moved the sentence and Figure 1 to the results section.
Line 83: Surgical procedures and rehabilitation plan-
Please elaborate regarding anesthesia. Were all patients undergoing the surgery under general anesthesia? Were there cases of regional anesthesia, as this may affect the early postoperative rehab.
Blade was used in all cases? or there were cases with a screw? please clarify.
Response: We thank the Reviewer for pointing out the variables affecting the early postoperative mobility.
All patients underwent surgery under general anesthesia. We then fixed the fracture with a Trochanteric Femoral Nail Advanced (TFNA) perforated “spiral blade” (DePuy Synthes, Paoli, PA), and not a screw type. We have edited the sentence as follows.
Page 3:
All patients underwent surgery under general anesthesia.
Line 93: Main exposure
Did you used a contrast material prior to cement injection to verify there is no leakage? Please elaborate.
Response: We did not use contrast material prior to cement injection. In surgery, we sufficiently confirmed that the guide pin of the lag screw did not perforate the femoral head bone using fluoroscopy.
Line 98:
We did not use contrast material to check the presence of perforation by guide pin insertion prior to cement injection. We sufficiently confirmed that the cement did not leak into the hip joint under fluoroscopy.
Line 108: “The secondary outcomes were pain scores measured using the visual analog scale (VAS) at rest and during movement on postoperative days 1–3.”
How can the VAS be attributed to the use of cement without considering the type of anesthesia, the number of analgesics consumed and other anesthesia parameters that can affect postoperative pain? Please explain or elaborate, if this was not assessed please include in the limitations of the study.
Response: We hypothesized that the initial stability early after the surgery would be improved by CA, which could be reflected by the reduction of mechanical pain as evaluated by the VAS.
We agree with Reviewer 1 that the postoperative pain can be also influenced by unmeasured anesthesia parameters, including the number of analgesics. Hence, we included this point as one of the limitations of the study.
Line 274:
Third, we did not evaluate the number of rescue analgesic used and the other anesthesia parameters for postoperative pain.
Results:
Line 159: “The study included 64 patients with a mean age of 87.0 (65–104) years. In total, 49 (77.8%) women and 14 (22.2%) men were included.”
49+14=63…
Response: We thank Reviewer 1 for pointing out our wrong description. We have edited it as follows.
Line 159:
The study included 63 patients with a mean age of 87.0 (65–104) years. In total, 49 (77.8%) women and 14 (22.2%) men were included
Table 1 – do you have an explanation why the average time from admission to surgery was more the 48 h?
Response: As pointed out, the recommended duration from admission to surgery in the world is within 48 hours [1,2]. However, in fact, the mean duration from admission to surgery of 3.3 days in this study is typical in Japan. The national inpatient database study in Japan reported that 77.5% of patients underwent surgery for hip fractures after 48 hours [3]. This is because in the Japanese clinical setting, the duration tends to be longer than the recommended 48 hours due to management of surgery teams including anesthesiologists.
[1] AAOS Updates Clinical Practice Guideline for Management of Hip Fractures in Older Adults 2021, https://www.aaos.org/aaos-home/newsroom/press-releases/aaos-updates-clinical-practice-guideline-for-management-of-hip-fractures-in-older-adults/
[2] Klestil T, Röder C, Stotter C, et al. Impact of timing of surgery in elderly hip fracture patients: a systematic review and meta-analysis. Sci Rep. 2018;8(1):13933.
[3] Sasabuchi Y, Matsui H, Lefor AK, Fushimi K, Yasunaga H. Timing of surgery for hip fractures in the elderly: A retrospective cohort study. Injury. 2018;49(10):1848-1854.
Were AO type 31A3 were excluded prior to the study? If so, add to the exclusion criteria.
Response: Our inclusion criteria intended to have all patients with AO/OTA 31A, including A3. As a result, we did not have any patients with A3 during our study period.
Line 189: “Patients with CA had higher Bis than those without”
A typo? Bis? Correct to BI
Response: Thank you for pointing out the error. We revised the sentence as follows.
Line 189:
Patients with CA had higher BI than those without, although the difference was not significant (p = 0.247).
Discussion:
Please comment, or discuss, why do you think, even though there is no difference in the VAS between groups, the cement still affects the early postoperative function?
Response: We thank Reviewer 1 for raising an important point. The causal relationship between CA and early postoperative function including pain remains unclear in our observational study.
Previous mechanical studies have primarily indicated that mechanical stability can be improved by CA. Our study results suggest that the improved mechanical stability achieved by CA may be also related to lower postoperative pain. The mechanisms of improved mechanical stability and functional outcomes (i.e., lower postoperative pain and higher mobility) need to be further investigated in future studies. We have now clarified this point in our Discussion.
Line 231:
We speculate that CA enhances initial mechanical stability, leading to improved ADL scores due to sufficient stability in the early postoperative period.
Please discuss the further cost of CA. for example, in the institution where I work, there is an allocation for a limited number of cements uses, for specific indications, since the cost of using cement is significantly higher than a CMN alone.
Response: We agree with your opinions. To date, no cost-effectiveness analysis (CEA) exists on this topic. We believe that CEA that considers the effect of CA should be conducted in the future. We have elaborated this point in the limitation of Discussion section as follows.
Line 271:
Fourth, we could not evaluate medical costs and clinical outcomes for a longer duration (3–12 months).
Limitations: Address the lack of reference to the type of anesthesia affects the early postoperative function
Response: We agree with your opinions. All patients underwent surgery under general anesthesia. We have added the sentences as follows.
Page 3:
All patients underwent surgery under general anesthesia.
Line 277: “Lastly, although there were several limitations, including unmeasured factors, in the present study, our results will draw attentions to the potential impact of CA on postoperative mobility. Ultimately, a well- designed study with a large sample size and long follow-up period is warranted to clarify the effectiveness of CA.”
This is not a limitation.
Response: As suggested, we have deleted the sentences.
Figures:
For the sake of the readers, I suggest adding a figure of a radiograph or intraoperative fluoroscopy that demonstrate the use CA.
Response: As suggested, we have added the figure as Figure 2.

Reviewer 2 Report
Dear Author,
thank you very much for submitting this interesting article.
It is a prospective cohort study.
Randomization did not take place in the study.
Aim study is to demonstrate the effectiveness of cement augmentation on the mobility of patients of the first 3 postoperative days. Study outcomes were collected between February and December 2021.
The primary outcome of the study is the "Cumulated Ambulation Score (CAS )" at postoperative days 1 to 3. CAS allows a valid assessment of the patient's mobility.
The secondary outcome of the study is the pain scale at rest and movement according to "Visual Analog Scale" (VAS).
The study included 64 patients divided into an experimental (n = 32) and control group (n = 32). No information is given about the whereabouts of the 64th patient.
The exact criteria used to assign patients to a group is not made clear.
Because the article is a prospective cohort study, its power is considered lower compared with randomized controlled trials. This can be illustrated by the following points:
Study type: prospective cohort study; no double blinding.
The classification of patients into experimental or control group is not randomized but done by the surgeons based on the indication criteria in Japan. The problem here is that the allocation is subjective and thus not reproducible.
Conclusion:
The mechanical stability and mobility of patients in the first 3 days is improved. However, increased CAS does not correlate with reduction in postoperative pain. The study design meets the requirements of a cohort study. Patients are divided into control and experimental groups according to risk factors, and criteria are established before the start of the study. Negative aspects that reduce the quality and significance of the study result from: Too small number of patients, subjective assignment to groups, no randomization, data are not collected on all patients (patient 64). The study shows an interesting question, but it needs further investigation and data collection.
In particular, however, it is critical to note that only 3 days of follow-up were performed. In my opinion, no valid statement can be made about the effectiveness of cement augmentation during this time. The pain was recorded one week postoperatively, why there is a difference between 3 days and one week is not quite clear.
In this respect, no effective statement can be made with this study.
Author Response
Response to Reviewer 2 Comments:
The study included 64 patients divided into an experimental (n = 32) and control group (n = 32). No information is given about the whereabouts of the 64th patient.The exact criteria used to assign patients to a group is not made clear.
Response: We thank Reviewer 2 very much for these comments. This study design is a prospective cohort study.
The indications for CA depend on the preference and judgment of the patient and surgeon and the inclusion and exclusion criteria are based on the standard protocols followed in Japan. The setting is similar to an actual clinical setting. We have edited the sentences as follows. 
Page 3:
The indications for CA depend on the preference and judgment of the patient and surgeon and are based on the indication criteria followed in Japan.
Because the article is a prospective cohort study, its power is considered lower compared with randomized controlled trials. This can be illustrated by the following points:
Study type: prospective cohort study; no double blinding.
The classification of patients into experimental or control group is not randomized but done by the surgeons based on the indication criteria in Japan. The problem here is that the allocation is subjective and thus not reproducible.
Response: As pointed out by the reviewer, this study is a prospective cohort study with some limitations; confounding by indication, subjective allocation. Further s randomized controlled trials should be conducted to confirm these results. We have added the sentence in the Discussion as follows.
Line 279:
Lastly, this study is a prospective cohort study with some limitations. Further well-designed randomized controlled trials are needed to clarify the effectiveness of CA.
Conclusion: The mechanical stability and mobility of patients in the first 3 days is improved. However, increased CAS does not correlate with reduction in postoperative pain. The study design meets the requirements of a cohort study. Patients are divided into control and experimental groups according to risk factors, and criteria are established before the start of the study. Negative aspects that reduce the quality and significance of the study result from: Too small number of patients, subjective assignment to groups, no randomization, data are not collected on all patients (patient 64). The study shows an interesting question, but it needs further investigation and data collection.
Response: As pointed out by the reviewer, our study had some limitations including its design and sample size. We have added the sentence as follows.
Line 279:
Lastly, this study is a prospective cohort study with some limitations. Further studies are needed to conduct well-designed randomized controlled trials to clarify the effectiveness of CA.
In particular, however, it is critical to note that only 3 days of follow-up were performed. In my opinion, no valid statement can be made about the effectiveness of cement augmentation during this time. The pain was recorded one week postoperatively, why there is a difference between 3 days and one week is not quite clear.
In this respect, no effective statement can be made with this study.
Response: As Reviewer 2 pointed out, the 1 week of follow up period as an outcome measurement was short. However, it is important to regain mobility as soon as possible for better mid- and long-term outcomes. We hypothesized that the mobility immediately after the surgery is largely influenced by postoperative pain. Therefore, we focused on this specific period (i.e., within 1 week of surgery) in this study. In addition, we tried to reduce some biases to enhance internal validity during this period.
As we described at the methods section, the pain scores were measured using the visual analog scale (VAS) at rest and during movement on postoperative days 1–3. The other following secondary outcomes were measured at postoperative one week; Barthel index, ambulance ability, return to the pre-fracture ambulatory level, postoperative complication classification system, and other medical complications.
We think that CAS is the specific measurement tool for very early postoperative periods such as postoperative 1–3 days [1,2]. On the other hand, Barthel index is a suitable measurement tool after achieving basic mobility around the bedside. We used these measurement tools at the appropriate period based on previous studies [1-4].
[1] Yamamoto N, Tomita Y, Ichinose A, et al. Cumulated ambulation score as predictor of postoperative mobility in patients with proximal femur fractures [published online ahead of print, 2022 Mar 15]. Arch Orthop Trauma Surg. 2022;10.1007/s00402-022-04401-9. doi:10.1007/s00402-022-04401-9
[2] Tomita Y, Yamamoto N, Inoue T, et al. Preoperative and perioperative factors are related to the early postoperative Barthel Index score in patients with trochanteric fracture. Int J Rehabil Res. 2022;45(2):154-160.
[3] Inui T, Watanabe Y, Kurata Y, et al. Early postoperative Barthel index score and long-term walking ability in patients with trochanteric fractures walking independently before injury: a retrospective cohort study. Arch Orthop Trauma Surg. 2021;141(8):1283-1290.
[4] Ogawa T, Aoki T, Shirasawa S. Effect of hip fracture surgery within 24 hours on short-term mobility. J Orthop Sci. 2019;24(3):469-473.

Round 2
Reviewer 1 Report
This study entitled “Effectiveness of cement augmentation on early postoperative mobility in patients treated for trochanteric fractures with cephalomedullary nailing: A prospective cohort study” is well written, and the authors should be applauded for their effort.
Title:
OK
Abstract:
Ok.
Introduction:
OK
Methods:
OK
Results:
OK
Discussion:
OK
Limitations:
OK
Figures:
OK
Author Response
We thank Reviewer 2 for the positive comments.